# DISSECTING MAHALANOBIS: HOW FEATURE GEOMETRY AND NORMALIZATION SHAPE OOD DETECTION

## ABSTRACT

Out-of-distribution (OOD) detection is critical for the reliable deployment and better understanding of deep learning models. To address this challenge, various methods relying on Mahalanobis distance were proposed and widely employed. However, the impact of representation geometry and feature normalization on the OOD performance of Mahalanobis-based methods is still not fully understood, which may limit their downstream application. To address this gap, we conducted a comprehensive empirical study across diverse image foundation models, datasets, and distance normalization schemes. First, our analysis shows that Mahalanobis-based methods aren't universally reliable. Second, we define the ideal geometry for data representations and demonstrate that spectral and intrinsic-dimensionality metrics can accurately predict a model's out-of-distribution (OOD) performance. Finally, we analyze how normalization impacts OOD performance. Building upon these studies, we propose a conformal generalization of recently proposed $\ell_2$ normalization that allows to control the degree of radial expansion of the representations geometry, which in turn helps improve OOD detection. By bridging the gap between representation geometry, normalization, and OOD performance, our findings offer new insights into the design of more effective and reliable deep learning models.

## 1 INTRODUCTION

Out-of-distribution (OOD) detection is foundational for building reliable, open-world vision systems, yet consistent evaluation at scale—especially with modern foundation models—remains challenging and essential for practice. Mahalanobis-based detectors (Lee et al., 2018) are surprisingly simple yet powerful baselines that often achieve state-of-the-art performance (Mueller & Hein, 2025; 2024). At its core, this approach models the feature distribution of in-distribution data—typically by fitting class-conditional multivariate Gaussians—and flags an input as OOD if its feature representation is far from all class centroids. While effective, it is not fully understood why this simple metric works so well or how the complex geometry of high-dimensional representations contributes to its success. This paper systematically investigates this question, revealing that representation geometry and feature normalization are the primary drivers of Mahalanobis-based OOD detection performance and providing a practical method to optimize them.

We begin our work by benchmarking a diverse set of self-supervised models, revealing significant variance in the inherent OOD detection capabilities of their representations. We then demonstrate that this variance is not random, but correlates strongly with measurable geometric properties of the in-distribution feature space, such as its intrinsic dimensionality and spectral structure. Unlike prior works Ren et al. (2021); Mueller & Hein (2025) focused on refining the distance metric itself, we introduce a direct control knob for the geometry of the representation: a $\beta$-scaled $\ell_2$ normalization that conformally contracts or expands the feature space. This allows us to reshape the representation to better suit the detector. Finally, we demonstrate

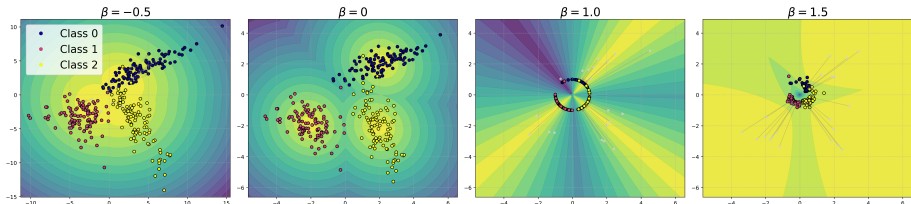

Figure 1: Effect of $\beta$-scaled $\ell_2$ normalization effect 2D feature geometry and Mahalanobis decision surfaces. Optimal $\beta$ improve OOD detection performance. Gray arrows indicate the mapping from the original to the transformed space. Increasing $\beta$ contracts norms and tightens clusters, yielding smoother, more localized decision regions; negative $\beta$ spreads points and broadens regions. Choosing an appropriate $\beta$ improves class separation and downstream OOD detection.

that a simple regression model can predict this optimal $\beta$, without access to target OOD data, achieving nearly the same performance as an oracle with access to OOD samples. Our main contributions are:

1. A comparative study of several Mahalanobis-based OOD detectors across diverse models, including a per-dimension analysis of its variants.
2. An analysis revealing that OOD performance is strongly predicted by the in-distribution geometry of features, such as spectral decay and intrinsic rank.
3. The introduction of $\beta$-scaled $\ell_2$ normalization and a method to predict the optimal $\beta$ using only in-distribution data.

## 2 RELATED WORK

Out-of-distribution (OOD) detection is essential for ensuring the reliability of machine learning systems in real-world deployment (Fort et al., 2021). Its goal is to identify whether inputs stem from the training distribution, thus preventing overconfident predictions on unexpected data (Yang et al., 2024). Post-hoc, training-free methods are particularly effective, as they combine efficiency with robustness without altering the model (Xu et al., 2023). Among OOD detection methods, Mahalanobis distance has become a cornerstone (Lee et al., 2018), with several refinements improving its robustness and performance. The standard Mahalanobis distance (MD) uses class-conditional covariance estimates to measure the distance of a sample from each class mean. In contrast, the Relative Mahalanobis distance (RMD) (Ren et al., 2021) compares each class-specific distance to a single global Gaussian fitted to all in-distribution (ID) data, effectively normalizing class distances against a global reference. Mahalanobis++ (Mueller & Hein, 2025) further improves performance by L2-normalizing features, making them adhere more closely to the Gaussian assumptions underlying the Mahalanobis distance. However, our sudy reveals broader insight on the influence of normalization while computing Mahalanobis distance, particularly in the context of vision models.

Vision OOD detection has shifted toward leveraging large-scale pretraining and contrastive objectives, where vision transformers Dosovitskiy et al. (2021) and CLIP Radford et al. (2021) show strong near-OOD performance and benefit markedly from few-shot outlier exposure and even label-only supervision for outlier classes (Fort et al., 2021). However, full fine-tuning can distort pretrained representations and harm OOD generalization relative to linear probing, with similar cautions for vision–language models; recent work also explores training-time scaling and post-hoc enhancements, and revisits detector design in vision foundation models (Fort et al., 2021; Ming & Li, 2024; Xu et al., 2023; Zhao et al., 2024b). Evaluation rigor has improved through ImageNet-scale suites like NINCO that mitigate in-distribution leakage, while theory and diagnostics connect feature separability to OOD error and delineate when OOD detection is learnable (Bitterwolf et al., 2023; XIE et al., 2023).

Representation geometry and normalization have attracted increased attention for their role in OOD generalization. Analyses of contrastive learning and normalization approaches (Le-Gia & Ahn, 2023; Tan et al., 2025) show that geometric priors, such as hyperspherical projection or $\ell_2$ normalization, can yield more robust representation spaces. Studies like (Zhao et al., 2024a) and (XIE et al., 2023) link improved feature separability and lower intrinsic dimensionality to higher OOD detection performance.

## 3 COMPARATIVE STUDY OF SELF-SUPERVISED MODELS

### 3.1 BACKGROUND

Let $z' = f(x') \in \mathbb{R}^d$ be the feature representation of a test image input $x'$, and let $\{\mathcal{N}(\mu_k, \Sigma)\}_{k=1}^K$ denote the $K$ class–conditional Gaussian distributions fitted to in–distribution (ID) training data.

**Mahalanobis distance (MD)** (Lee et al., 2018) measures the squared distance of $z'$ from the mean of each class: $\mathrm{MD}_k(z') = (z' - \mu_k)^\top \Sigma^{-1} (z' - \mu_k)$. A confidence score is obtained as the negative minimum distance, $\mathcal{C}(x') = -\min_k \mathrm{MD}_k(z')$.

**Relative Mahalanobis distance (RMD)** (Ren et al., 2021) compares the class-specific distance to a single global Gaussian fitted to all ID data, the **marginal Mahalanobis distance (MMD)**: $\mathrm{MD}_0(z') = (z' - \mu_0)^\top \Sigma_0^{-1} (z' - \mu_0)$, and defines the RMD score as: $\mathrm{RMD}_k(z') = \mathrm{MD}_k(z') - \mathrm{MD}_0(z')$. This effectively normalizes class distances against a global reference.

**Eigenvalue Decomposition.** Let $\Sigma = U\Lambda U^\top$ be the eigendecomposition of the shared covariance matrix, where $\Lambda = \mathrm{diag}(\lambda_1, \ldots, \lambda_d)$ and $U$ is orthonormal. As shown in (Mueller & Hein, 2025), in this basis the regular Mahalanobis distance decomposes as

$$\mathrm{MD}_k(z') = \sum_{i=1}^d \frac{\left[u_i^\top (z' - \mu_k)\right]^2}{\lambda_i}, \tag{1}$$

revealing the contribution of each principal component. The eigenvalues $\{\lambda_i\}$ quantify the spread of in-distribution features along each direction; small $\lambda_i$ correspond to directions of small variance and thus high discriminative power for OOD detection.

### 3.2 CROSS-MODEL OOD DETECTION PERFORMANCE

Understanding how different representation learning strategies affect OOD detection is a critical first step in our study. Mahalanobis-based detectors are widely used, but their sensitivity to model architecture, pretraining data, and fine-tuning is not well characterized. We therefore begin with a broad, model-agnostic comparison to answer a simple question: *Which modern self-supervised or pretrained vision models produce representations that naturally lend themselves to Mahalanobis-style OOD detection?*

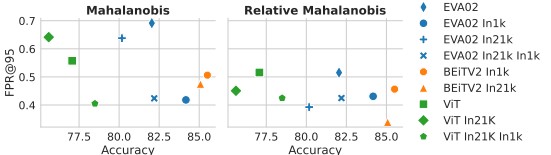

Figure 2: OOD detection performance across model families on the NINCO. RMD consistently outperforms the standard MD, especially for models pretrained but not fine-tuned on ImageNet.

To this end, we gathered publicly available checkpoints from `timm` (Wightman, 2019) and `huggingface-transformers` (Wolf et al., 2020), spanning a range of architectures, sizes, and pretraining objectives. Following the OpenOOD protocol (Yang et al., 2022), we evaluate on five standard benchmarks—NINCO (Bitterwolf et al., 2023), iNaturalist (Van Horn et al., 2018), SSB-Hard (Bitterwolf

et al., 2023), OpenImages-O (Krasin et al., 2017), and Textures (Cimpoi et al., 2014). Performance is reported as the false positive rate at 95% true positive rate (FPR@95 or FPR). Additional implementation details and a full model list appear in the Appendix.

**Key Insights (Figure 2)** *(1) RMD consistently improves performance* over standard Mahalanobis distance, with the largest gains in models pretrained—but not fine-tuned—on ImageNet. For example, RMD markedly boosts the OOD detection of EVA02-In21k and ViT-In21k, matching or surpassing their fine-tuned counterparts. This weakens the usual correlation between in-distribution accuracy and FPR and yields more uniform score distributions. *(2) Classification accuracy is not a reliable proxy for OOD performance.* Substantial accuracy gaps (often $> 10\%$) do not necessarily translate into improved detection, though we observe a mild correlation along the fine-tuning sequence In1k $\rightarrow$ In22k-In1k $\rightarrow$ large In22k-In1k models (complete results in Appendix D.1).

### 3.3 MAHALANOBIS VARIANTS AND PER-DIMENSION ANALYSIS

Having established cross-model trends, we next ask: *Which aspects of the Mahalanobis representation space actually drive OOD discrimination?* Beyond aggregate scores, the structure of individual feature dimensions may reveal why certain models excel while others falter. We therefore conduct a detailed *per-dimension* investigation using three Mahalanobis variants—regular, marginal, and relative. Using the decomposition in Eq. equation 1, we define the *OOD separation* of the $i$-th eigenvector direction as the difference between its mean contribution for out-of-distribution samples and for in-distribution samples:

$$S_i = \mathbb{E}_{x' \sim \mathcal{D}_{\text{OOD}}} \left[ \frac{\left[ u_i^\top (z' - \mu_k) \right]^2}{\lambda_i} \right] - \mathbb{E}_{x \sim \mathcal{D}_{\text{ID}}} \left[ \frac{\left[ u_i^\top (z - \mu_k) \right]^2}{\lambda_i} \right].$$ Here $S_i$ quantifies how strongly dimension $i$

separates OOD from in-distribution data; positive values indicate greater OOD spread along that eigendirection. The eigenvalues $\{\lambda_i\}$ are sorted in descending order ($\lambda_1 \geq \lambda_2 \geq \cdots \geq \lambda_d$), so dimension $i$ on subsequent plots corresponds to the $i$-th largest eigenvalue.

Figure 3 presents two complementary analyses. The top row reports **OOD separation**, exposing which latent directions contribute most to detection. The bottom row shows a **dimension-ablation** study: we incrementally compute FPR using the first $K$ principal components (forward) or start from the least-variant dimensions (backward).

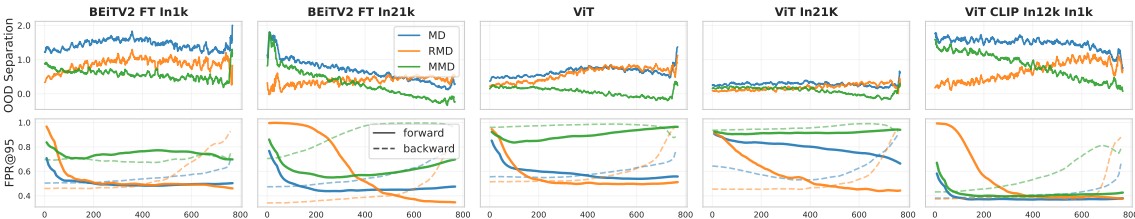

Figure 3: Dimension-wise analysis of OOD separation (top) and FPR under progressive dimension ablation (bottom). Large embedding-space separation does not necessarily guarantee superior detection.

**Key Insights** *(1) Large OOD separation $S_i$ do not always yield lower FPR:* for example, BEiTV2 FT In1k exhibits stronger separation across all three distance metrics yet performs on par with—or worse than—BEiTV2 FT In21k. *(2) The number of dimensions required for optimal detection varies widely:* some models saturate quickly, whereas others need nearly the full spectrum. Backward ablation reveals that ViT In21k achieves its best FPR using only the second half of the spectrum, indicating that directions of *smaller explained variance* can be more discriminative for OOD-ness. *(3) MMD results show that certain models*

*rely heavily on class-discriminative features*, while others—such as CLIP—spread OOD samples far from ID data regardless of their classes, consistently achieving low MMD scores.

# 4 GEOMETRY OF REPRESENTATIONS

In the previous section we showed that no single OOD method yield consistent performance and behavior across multiple models. In fact, different SSL models and pretraining regimes produce representations with distinct geometric properties, indicating that OOD performance depends on the intrinsic structure of the representation space. To understand these effects, we analyze the **internal geometry of model representations**, seeking to answer: *What internal characteristics of a model's feature space predict strong OOD detection, and how do pretraining and fine-tuning shape these characteristics?*

## 4.1 SPECTRAL ANALYSIS

To understand how the intrinsic geometry of representations affects OOD performance, we begin by examining the spectral properties of three key matrices: the feature covariance $C$, the within-class scatter $S_w$, and the between-class scatter $S_b$. These matrices capture complementary aspects of the feature space: $C$ reflects overall variance, $S_w$ measures intra-class dispersion, and $S_b$ quantifies inter-class separation (more details in Appendix A). Our first analysis focuses on the eigenvalue spectra of these matrices. The magnitude and decay of eigenvalues reveal how variance is distributed across dimensions, providing insight into the richness and anisotropy of the feature space. For instance, a steep decay in $S_w$ eigenvalues indicates that intra-class variability is concentrated along a few directions, resulting in tight clusters, whereas a slower decay suggests more diffuse intra-class variation. Similarly, large eigenvalues in $S_b$ correspond to well-separated class means, signaling strong discriminability.

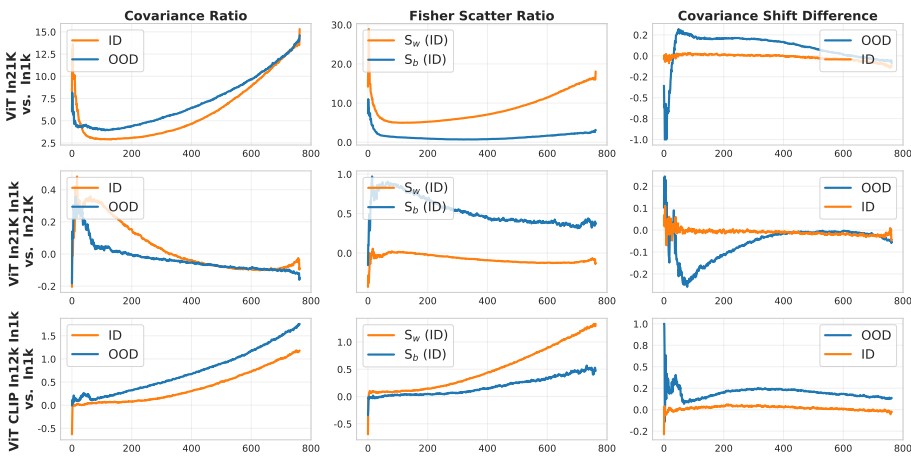

Figure 4: Spectral ratios across models. Higher ratios indicate richer within-class variation and more expressive feature spaces. Fine-tuning tends to increase $S_b$ while preserving $S_w$.

**Spectral ratios**  To systematically compare models, we compute ratios between eigenvalues of $S_b$, $S_w$, and $C$. These ratios serve as compact summaries of representation geometry. Higher $S_b/S_w$ ratios indicate representations with greater between-class separation relative to intra-class spread, which generally favors OOD detection, while lower ratios may signal overlapping clusters or limited discriminative power. A higher $C$ ratio indicates that variance is distributed along multiple directions, reflecting a richer and more expressive

representation that can better accommodate novel OOD inputs without major distortion. As illustrated in Figure 4, models pretrained on large, diverse datasets (e.g., In21k) exhibit larger $C$ and $S_w$ ratios, capturing richer intra-class variations and producing more expressive feature spaces. Fine-tuning tends to increase $S_b$ ratios while preserving $S_w$, enhancing class separability without sacrificing cluster compactness. Models trained on smaller datasets exhibit smaller ratios, reflecting less expressive representations with weaker discriminability.

**Eigenvalue shifts**  Beyond static spectra, we are interested in how stable the representation geometry is under distributional shifts. To capture this, we define a *spectral shift metric*, which measures the relative change in eigenvalues from the training set to validation or OOD data (see Appendix A). A small shift indicates that the representation preserves its structure across data splits, signaling robustness. Large positive shifts reveal that features are spreading along new directions, while large negative shifts indicate compression. Figure 4 shows that OOD samples induce larger spectral shifts in models trained on small datasets, reflecting lower generalization and brittle feature structures. Large-scale pretrained models show smaller shifts, indicating more stable, robust representations under distributional change. Fine-tuning generally maintains small shifts while increasing $S_b$, improving class separation without compromising intra-class compactness.

## 4.2 GEOMETRIC TRADE-OFFS

To systematically identify what makes a representation "good" for OOD detection, we correlate spectral and manifold-based metrics with detection performance. The geometry of the representations is assessed with two complementary families of measures: **manifold-geometry metrics**—including *Intrinsic Dimensionality (ID)* and *Local Intrinsic Dimensionality (LID)* (Ma et al., 2018)—and **eigenvalue-based metrics** (e.g., *Entropy*, *Slope*, *Fisher Ratio*) computed on the covariance matrix $C$ and the Fisher scatter matrices $S_w$ and $S_b$. A complete description of all metrics is provided in Appendix B. Correlation analysis, visualized in Figure 5, highlights several metrics that correlate strongly with OOD performance.

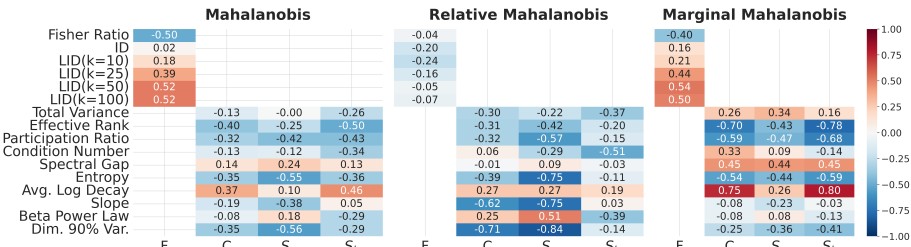

Figure 5: Spearman correlations between representation metrics and OOD performance across Mahalanobis variants. The three Mahalanobis-based detectors exploit different geometric cues, which explains their distinct correlation patterns.

**Key insights**  *Mahalanobis distance* shows moderate correlation with individual metrics; its sensitivity to the global eigen-spectrum can make it less reliable for ambiguous OOD regions. *Relative Mahalanobis* emphasizes how well features fit their class cluster by normalizing out global variance. It correlates strongly with $S_w$ metrics, such as entropy and eigenvalue decay, reflecting the importance of compact, well-separated clusters. *Marginal Mahalanobis* ignores class structure and correlates primarily with global metrics ($C$ and $S_b$), indicating that its success depends on overall manifold shape rather than per-class separation.

### 4.3 IDEAL GEOMETRY

OOD detection performance depends on both the magnitude of separation between ID and OOD features and the internal structure of the representation. Ideally, an effective representation exhibits a balance between local manifold complexity and intra-class compactness: low-dimensional manifolds require tighter clusters to separate OOD data, while high-dimensional manifolds allow looser clusters, as extra directions naturally push OOD samples away. This "ideal geometry" reflects a compensatory relationship between these two principles. However, as we observed in Sections 3.3 and 4.2, single metrics fail to satisfy both principles simultaneously. Models with larger raw feature separation can perform worse than models with smaller separation, indicating that additional geometric properties beyond simple separation are critical for reliable detection. While individual metrics provide insight, they capture only a single aspect of the geometry.

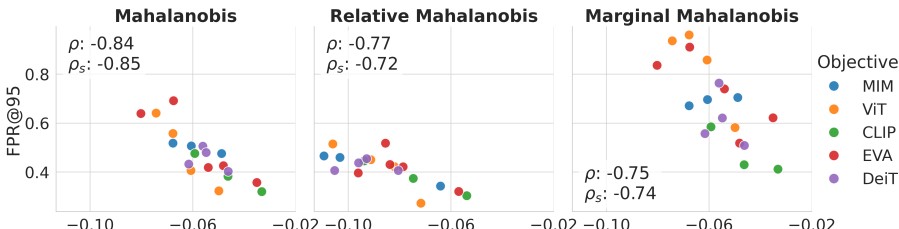

Figure 6: Correlation of the product of LID and $S_w$ slope with OOD detection performance. Minimizing the absolute value indicates an optimal balance between manifold richness and cluster compactness.

To address this limitation, we consider combinations of metrics across different geometric aspects. In particular, the product of LID and the slope of $S_w$ strongly predicts OOD performance, which is shown in Figure 6. For the standard MD, this combined metric achieves a Spearman correlation of $\rho_s = 0.85$, and it remains highly predictive for RMD and MMD ($\rho_s = 0.72$ and $0.74$, respectively).

In practice, when the feature manifold is locally simple (low *LID*), the model has fewer dimensions to separate ID and OOD data, requiring extremely tight and compact class clusters (a steep, negative *slope*). The detector thus relies on strict confinement of ID data to identify outliers. Conversely, when the manifold is locally rich and complex (high *LID*), the space itself helps isolate OOD samples, allowing less compact clusters (a shallower *slope*), as high dimensionality drives OOD separation. The product of LID and slope reaches its ideal when minimized in absolute magnitude, reflecting the optimal balance between local manifold richness and cluster compactness.

## 5 NORMALIZATION EFFECTS

Our preceding analysis has established that the efficacy of Mahalanobis-based OOD detectors is intrinsically linked to the geometric structure of a model's feature space. Yet, even when the geometry is favorable, standard Mahalanobis distance assumes Gaussian-distributed features with tied covariance, an assumption often violated in practice. Empirical studies, such as Mahalanobis++ (Mueller & Hein, 2025), show that deep neural network features frequently exhibit heavy-tailed distributions and large variations in feature norms. These deviations imply that ID features do not occupy a single, globally Euclidean space; instead, they lie on a collection of low-dimensional, non-Euclidean submanifolds. Consequently, the raw Mahalanobis metric can misestimate distances, undermining OOD detection. These observations motivate explicit control over feature geometry: by adjusting feature magnitudes we can either better satisfy the Mahalanobis assumptions or deliberately reshape the space to improve OOD separation.

Table 1: False-positive rate (FPR) across models (averaged over datasets) using different Mahalanobis variants: **MD\*** uses the empirically optimal $\beta$, $\hat{\textbf{MD}}$ uses the regression-predicted $\hat{\beta}$, **MD** (standard) fixes $\beta = 0$, and **MD++** (Mahalanobis++) fixes $\beta = 1$. The regression-guided detector generally outperforms fixed-$\beta$ settings.

| Detector | BEiTV2 In1k | BEiTV2 In21k | ViT | ViT In21K | ViT In21K In1k | ViT-L In21K In1k | DeiT3 | DeiT3 In21k In1k | DeiT3-L In22k In1k | EVA02 | EVA02 In21k | EVA02 In21k In1k | EVA02-L In22k In1k | ViT CLIP In1k | ViT CLIP In12k In1k | Average |
|---|---|---|---|---|---|---|---|---|---|---|---|---|---|---|---|---|
| MD* | 0.365 | 0.244 | 0.445 | 0.493 | 0.327 | 0.229 | 0.425 | 0.348 | 0.320 | 0.378 | 0.468 | 0.369 | 0.351 | 0.364 | 0.246 | 0.358 |
| $\hat{\text{MD}}$ | 0.375 | 0.274 | 0.456 | 0.518 | 0.346 | 0.239 | 0.428 | 0.356 | 0.333 | 0.401 | 0.498 | 0.383 | 0.368 | 0.386 | 0.254 | 0.375 |
| MD | 0.402 | 0.436 | 0.457 | 0.532 | 0.357 | 0.253 | 0.433 | 0.376 | 0.366 | 0.534 | 0.565 | 0.408 | 0.370 | 0.402 | 0.334 | 0.415 |
| MD++ | 0.376 | 0.298 | 0.454 | 0.577 | 0.387 | 0.282 | 0.430 | 0.356 | 0.342 | 0.446 | 0.506 | 0.382 | 0.378 | 0.382 | 0.278 | 0.392 |
| RMD* | 0.363 | 0.312 | 0.442 | 0.378 | 0.365 | 0.259 | 0.386 | 0.331 | 0.335 | 0.426 | 0.358 | 0.373 | 0.308 | 0.366 | 0.294 | 0.353 |
| RMD | 0.367 | 0.324 | 0.443 | 0.392 | 0.375 | 0.268 | 0.390 | 0.338 | 0.347 | 0.442 | 0.368 | 0.383 | 0.314 | 0.370 | 0.297 | 0.361 |
| RMD | 0.391 | 0.331 | 0.449 | 0.398 | 0.376 | 0.269 | 0.408 | 0.366 | 0.376 | 0.440 | 0.367 | 0.403 | 0.326 | 0.403 | 0.325 | 0.375 |
| RMD++ | 0.373 | 0.325 | 0.446 | 0.398 | 0.376 | 0.269 | 0.399 | 0.351 | 0.359 | 0.440 | 0.377 | 0.391 | 0.318 | 0.386 | 0.309 | 0.368 |

## 5.1 Conformal $\ell_2$ Normalization

To control the geometry of the representation space and mitigate the sensitivity to raw feature norms, we introduce a **conformal $\ell_2$ normalization** that applies a radially symmetric conformal map to each feature vector. For a feature $z \in \mathbb{R}^d \setminus \{0\}$, the transformation is defined as:

$$\phi_\beta(z') = \frac{z'}{\|z'\|^\beta}, \qquad (2)$$

where the scalar parameter $\beta \in \mathbb{R}$ governs the amount of **radial scaling**. This mapping is **angle-preserving** (conformal) and induces the Riemannian metric $g_\beta = \|z'\|^{-2\beta} g_{\text{Euc}}$, where distances in the transformed space are measured with respect to a conformally rescaled Euclidean metric. The new radius of the feature is given by $\|\phi_\beta(z)\| = \|z\|^{1-\beta}$ (see Figure 1 and Appendix C for details).

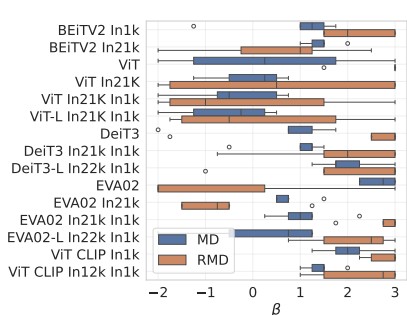

Figure 7: Distribution of empirically optimal $\beta$ for MD and RMD detectors across OOD datasets. High variability indicates model- and dataset-specific tuning is needed.

Intuitively, adjusting $\beta$ allows us to compress or expand the feature space radially, controlling how tightly features cluster near the origin or how much "empty space" surrounds them:

- $\beta < 0$: Radii are expanded, aggressively emphasizing differences between norms.
- $\beta = 0$: Radii are unchanged, corresponding to the standard Mahalanobis distance.
- $0 < \beta < 1$: Radii are compressed, while preserving relative ordering.
- $\beta = 1$: All points are projected onto the unit sphere, equivalent to the MD++ normalization.
- $\beta > 1$: Features farther from the origin are pulled inward more strongly than those closer, creating strong radial contraction.

By tuning $\beta$, we obtain a flexible mechanism to shape the feature geometry, providing a simple yet powerful knob for improving OOD separation while maintaining compact ID clusters.

## 5.2 Predicting the Optimal $\beta$

Our experiments show that the optimal conformal parameter $\beta$ is highly model- and dataset-dependent, reflecting the intrinsic geometry of the learned representations. Moderate positive values often align the feature distribution with the Gaussian, tied-covariance assumptions of the Mahalanobis detector, yet in some cases larger or even negative values yield stronger in/out-of-distribution separation. Consequently, a fixed

choice of $\beta$ is rarely optimal. Figure 7 shows the empirically optimal $\beta$ values (searched over $[-2, 3]$ in 0.25 steps) for MD and RMD detectors across different OOD datasets. The wide spread of optimal values underscores that a one-size-fits-all approach is ineffective.

**Regression Framework.** To eliminate the need for tuning on the target OOD dataset, we train a regression model that predicts $\beta$ using in-distribution geometry metrics, while allowing it to learn from other OOD datasets. We adopt a Leave-One-Dataset-Out scheme: for each target OOD dataset, the regression model is trained on all other OOD datasets and their corresponding ID features, ensuring that the target OOD samples are never seen during training. This setup is conceptually similar to outlier exposure (Hendrycks et al., 2018), where access to auxiliary OOD data helps guide the detector, but crucially, here the model generalizes to completely unseen OOD distributions. Candidate predictors include spectral properties of the feature covariance, intrinsic dimensionality estimates, and other representation-geometry statistics; highly collinear features ($\rho > 0.9$) are removed. The target variable is the $\beta$ value minimizing the false-positive rate (FPR) for each model–dataset pair. Let $\hat{\beta}$ denote the predicted value.

**Results.** Table 1 reports FPR across models and datasets. The regression-predicted $\hat{\beta}$ consistently improves OOD detection compared to fixed baselines ($\beta = 0$ for standard MD, $\beta = 1$ for MD++), for both MD and RMD detectors. For MD, the regression achieves a MAE of 0.72 and $R^2 = 0.25$ in predicting the optimal $\beta$; for RMD, the MAE is 0.89 with $R^2 = 0.47$. While the regression does not perfectly recover the empirically optimal $\beta$, it captures sufficient geometric information from in-distribution features to meaningfully improve detection. Figure 8 visualizes predicted vs. optimal $\beta$ for MD. Points close

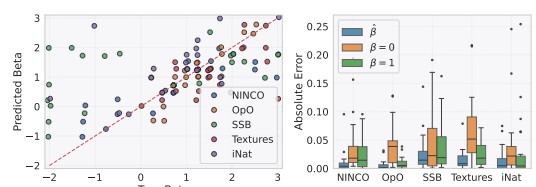

Figure 8: Predicted vs. optimal $\beta$ for MD under Leave-One-Dataset-Out validation. The diagonal indicates perfect prediction.

to the diagonal indicate that the regression captures key trends even for previously unseen OOD datasets. This demonstrates a practical path for tuning conformal $\ell_2$ scaling without access to target OOD data, leveraging ID feature structure and prior knowledge from other auxiliary OOD datasets.

# 6 CONCLUSION

In this work, we conducted a comprehensive empirical study across diverse image foundation models, datasets, and distance normalization schemes to understand how representation geometry shape Mahalanobis-based OOD detection performance. Our comparative analysis revealed that these detectors are not universally reliable, with significant variance in inherent OOD detection capabilities across different self-supervised models. We demonstrated that this variance correlates strongly with measurable geometric properties of the in-distribution feature space. In particular, the product of Local Intrinsic Dimensionality (LID) and within-class scatter slope achieves strong correlations with detection effectiveness. The proposed measure reflects the balance between local manifold complexity and cluster compactness, crucial for effective OOD.

Building on these geometric insights, we introduced $\beta$-scaled $\ell_2$ normalization, a conformal transformation that enables direct control over radial geometry, allowing practitioners to reshape feature spaces to better align with Mahalanobis assumptions or enhance OOD separation. We developed a regression framework that successfully predicts optimal $\beta$ values using only in-distribution training data, achieving performance on-par with the oracle. Future work should focus on developing fully model-free methods for determining optimal $\beta$.

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

## A    COMPUTATION AND INTUITION FOR COVARIANCE AND SCATTER MATRICES

Our spectral analysis relies on three core matrices derived from the feature embeddings $\{z_i, y_i\}_{i=1}^N$ of a trained model: the overall feature covariance $C$, the within-class scatter $S_w$, and the between-class scatter $S_b$. All eigenvalues reported in the main text are sorted in descending order.

**Feature Covariance.**    The global covariance is

$$C = \frac{1}{N} \sum_{i=1}^N (z_i - \mu)(z_i - \mu)^\top, \qquad \mu = \frac{1}{N} \sum_{i=1}^N z_i. \tag{3}$$

This matrix captures the overall spread of representations in feature space. Large eigenvalues correspond to directions of high variance, indicating axes along which the model representation varies the most across all samples.

**Within-Class Scatter.**    For $K$ classes with means $\mu_k$ and $n_k$ samples each,

$$S_w = \frac{1}{N} \sum_{k=1}^K \sum_{i:y_i=k} (z_i - \mu_k)(z_i - \mu_k)^\top. \tag{4}$$

$S_w$ measures the average dispersion of features around their class means, capturing intra-class variability. Intuitively, if $S_w$ has large eigenvalues, samples of a class are more spread out in feature space; small eigenvalues indicate tight, compact clusters. Notably, $S_w$ is *equivalent to the tied covariance* $\Sigma$ used in Mahalanobis distance (Eq. 1) when all classes share a common covariance estimate: $\Sigma = S_w$. Thus the Mahalanobis detector implicitly measures distances with respect to the within-class scatter of the training distribution.

**Between-Class Scatter.**    The between-class scatter quantifies the variability of class means:

$$S_b = \frac{1}{N} \sum_{k=1}^K n_k (\mu_k - \mu)(\mu_k - \mu)^\top. \tag{5}$$

This matrix captures inter-class separation: directions with large eigenvalues indicate axes along which class centroids are widely separated, while small eigenvalues correspond to directions where classes overlap. The spectrum of $S_b$ provides insight into the model's ability to linearly discriminate between classes.

**Spectral Shift Metric.**    To study how representations change under distributional shifts, for each matrix $M \in \{C, S_w, S_b\}$ we compute its eigenvalues $\{\lambda_i^{\text{train}}\}$ on the training set and $\{\lambda_i^{\text{eval}}\}$ on a validation or OOD set. The relative eigenvalue shift is defined as

$$\Delta_i(M) = \frac{\lambda_i^{\text{eval}} - \lambda_i^{\text{train}}}{\lambda_i^{\text{train}}}. \tag{6}$$

This spectrum of shifts highlights how the geometry of the representation changes under distributional shift, providing a fine-grained indicator of robustness or overfitting.

**Intuition Behind the Shift Metric.**

- **Zero shift ($\Delta_i \approx 0$):** The corresponding direction in feature space is stable across data splits.
- **Positive shift ($\Delta_i > 0$):** The representation spreads out along this eigenvector in the new data, increasing variance.
- **Negative shift ($\Delta_i < 0$):** The representation compresses along this eigenvector, reducing variance.
- **Magnitude:** Reflects the relative degree of expansion or contraction. For example, $\Delta_i = 0.5$ indicates a 50% increase in variance, while $\Delta_i = -0.2$ indicates a 20% decrease.

**Interpretation in Model Analysis.**

- **Small shifts across all eigenvectors:** Robust and stable representations that generalize well.
- **Large positive shifts:** Features become more variable on new data, potentially indicating under-regularization or sensitivity to OOD inputs.
- **Large negative shifts:** Features compress on new data, potentially indicating overfitting.
- **Consistent shift patterns:** Systematic changes in representation geometry, revealing overfitting or robustness issues.

**Types of Shifts.**

- **Validation covariance shift:** Change in global covariance from training to validation data.
- **OOD covariance shift:** Change in global covariance from training to out-of-distribution data.
- **Validation within-class shift:** Change in within-class scatter from training to validation data.
- **Validation between-class shift:** Change in between-class scatter from training to validation data.

## B    DETAILED DESCRIPTION OF SPECTRAL AND MANIFOLD METRICS

Let $X \in \mathbb{R}^{N \times d}$ be the feature matrix and $C = \frac{1}{N}(X - \bar{X})^{\top}(X - \bar{X})$ its covariance. All metrics below operate on the sorted non-negative eigenvalues of $C$, denoted $\lambda_1 \geq \lambda_2 \geq \cdots \geq \lambda_d$.

**Intrinsic Dimensionality (ID).**    Global estimate of the manifold dimension using maximum-likelihood methods of Ma et al. (2018).

**Local Intrinsic Dimensionality (LID).**    For a point $z$,

$$\mathrm{LID}_k(z) = -\left[ \frac{1}{k} \sum_{j=1}^{k} \log \frac{r_j(z)}{r_k(z)} \right]^{-1},$$

where $r_j$ is the $j$-th nearest-neighbor distance. We report the dataset mean for $k \in \{10, 25, 50, 100\}$.

**Total Variance.**

$$\text{Total Variance} = \sum_{i=1}^{d} \lambda_i.$$

**Effective Rank.**    Implements the code's definition,

$$\text{Effective Rank} = \frac{\sum_i \lambda_i}{\lambda_1},$$

the ratio of total variance to the largest eigenvalue.

**Participation Ratio.**

$$\text{PR} = \frac{\left(\sum_i \lambda_i\right)^2}{\sum_i \lambda_i^2}.$$

**Condition Number.**

$$\kappa = \frac{\lambda_1}{\lambda_d},$$

where $\lambda_d$ is the smallest positive eigenvalue.

**Spectral Gap.**    Difference between the largest and *sixth* largest eigenvalue,

$$\text{Gap} = \lambda_1 - \lambda_6.$$

**Entropy.**    Shannon entropy of the normalized spectrum,

$$\text{Entropy} = -\sum_i p_i \log p_i, \qquad p_i = \frac{\lambda_i}{\sum_j \lambda_j}.$$

**Average Log Decay Rate (Top-20).**    Mean forward difference of the first 20 log-eigenvalues,

$$\frac{1}{19} \sum_{i=1}^{19} \left[\log \lambda_i - \log \lambda_{i+1}\right].$$

**Slope.**    Slope of the least-squares fit

$$\log \lambda_i = a + b\, i,$$

i.e., regression of $\log \lambda_i$ on the linear index $i$.

**Beta Power Law.**    Exponent $\beta$ of a power law $\lambda_i \propto i^{-\beta}$, computed as the negative slope of $\log \lambda_i$ versus $\log i$.

**Dim. 90% Var.**    Minimum $k$ such that $\sum_{i=1}^{k} \lambda_i \big/ \sum_j \lambda_j \geq 0.9$.

These definitions ensure exact reproducibility of the results reported in Section 4.3.

## C    GEOMETRY INDUCED BY CONFORMAL $\ell_2$ NORMALIZATION

This appendix provides a formal derivation of the Riemannian metric naturally associated with the conformal $\ell_2$ normalization used in our Mahalanobis-based OOD detector.

## C.1 SETUP AND DERIVATION

Let $z \in \mathbb{R}^d \setminus \{0\}$ be a feature vector with the standard Euclidean metric $g_{\text{Euc}}$. We apply a radially symmetric conformal map, $\phi_\beta$, to the feature space:

$$\phi_\beta(z) = \frac{z}{\|z\|^\beta},$$

where the scalar parameter $\beta \in \mathbb{R}$ controls the degree of radial scaling. This smooth map induces a new Riemannian metric on the domain via the *pullback* of the Euclidean metric, $g_\beta := \phi_\beta^* g_{\text{Euc}}$.

The differential of $\phi_\beta$ is given by

$$\mathrm{d}\phi_\beta(z) = \|z\|^{-\beta} \left[ I - \beta \frac{zz^\top}{\|z\|^2} \right].$$

Using this, we can derive the induced metric $g_\beta$ by computing the pullback of the Euclidean metric. For a tangent vector $v$, the squared norm in the new metric is $g_\beta(v, v) = v^\top \mathrm{d}\phi_\beta(z)^\top \mathrm{d}\phi_\beta(z) v$.

To simplify this expression, we decompose any tangent vector $v$ into a radial component $v_r$ and an angular component $v_\perp$ in polar coordinates $z = ru$ ($r = \|z\|$, $u = z/r$). This radial-angular decomposition reveals the structure of the induced metric:

$$\boxed{g_\beta = r^{-2\beta} \, \mathrm{d}r^2 + r^{2(1-\beta)} g_{S^{d-1}},}$$

where $g_{S^{d-1}}$ is the standard round metric on the unit sphere. Thus $\beta$ continuously interpolates between Euclidean geometry ($\beta = 0$), spherical contraction ($\beta > 0$), and radial expansion ($\beta < 0$).

## C.2 GEOMETRIC INTERPRETATION OF THE PARAMETER $\beta$

The exponent $\beta$ is a single parameter that continuously interpolates between different geometries by controlling the radial-tangential trade-off.

- **Case 1: $\beta = 0$ (Euclidean Geometry).** The conformal map reduces to the identity, and the induced metric becomes the standard Euclidean metric ($r^0 \mathrm{d}r^2 + r^2 g_{S^{d-1}}$).
- **Case 2: $\beta > 0$ (Contractive Geometries).** This mapping pulls points towards the unit hypersphere. The new radius becomes $\|z\|^{1-\beta}$. If $0 < \beta < 1$, radii are compressed but their order is preserved. For $\beta = 1$ we obtain a hypersphere, corresponding to standard $\ell_2$ normalization. When $\beta > 1$, points that were far from the origin are pulled inward even more strongly, which can be seen as inducing a hyperbolic-like geometry. This contractive effect makes the ID distribution closer to the Gaussian assumptions of the Mahalanobis detector.
- **Case 3: $\beta < 0$ (Expansive Geometries).** Let $\beta = -\gamma$ for $\gamma > 0$. The mapping becomes $\phi_{-\gamma}(z) = z \cdot \|z\|^\gamma$. This transformation pushes points with large norms even further from the origin, radially stretching the space. This is optimal when ID class manifolds are already well-separated and situated far from the origin, as it further increases inter-cluster distances while creating a large, empty void around the origin where OOD samples can be easily detected.

## C.3 CONFORMAL MAHALANOBIS DISTANCE

The final OOD detector combines this geometric transformation with the standard Mahalanobis distance calculation. After applying a whitening transformation to the data, we apply the conformal map $\phi_\beta$ and then compute the standard Mahalanobis distance between the mapped test feature and the mapped class mean. The final OOD score is the minimum distance to any class mean:

$$\text{Score}(z) = \min_c D_\beta(z, \mu_c).$$

By tuning $\beta$ we directly control this geometry, providing a principled way to align—or deliberately mis-align—the feature space with the statistical assumptions underlying Mahalanobis-based OOD detection.

# D  FULL RESULTS

## D.1  CROSS-MODEL PERFORMANCE

Table 2: False positive rate across different models and datasets using two Mahalanobis distance variants (MD and RMD)

| Model | Mahalanobis | | | | | | Relative Mahalanobis | | | | | |
|---|---|---|---|---|---|---|---|---|---|---|---|---|
| | NINCO | OpO | SSB | Textures | iNat | Average | NINCO | OpO | SSB | Textures | iNat | Average |
| BEiTV2 In1k | 0.506 | 0.212 | 0.825 | 0.327 | 0.142 | 0.402 | 0.456 | 0.207 | 0.827 | 0.323 | 0.140 | 0.391 |
| BEiTV2 In21k | 0.475 | 0.331 | 0.781 | 0.418 | 0.174 | 0.436 | 0.339 | 0.165 | 0.786 | 0.314 | 0.051 | 0.331 |
| MAE In1k | 0.516 | 0.272 | 0.827 | 0.353 | 0.208 | 0.435 | 0.463 | 0.251 | 0.828 | 0.360 | 0.186 | 0.418 |
| DINOV2 | 0.424 | 0.178 | 0.773 | 0.302 | 0.014 | 0.338 | 0.577 | 0.200 | 0.845 | 0.415 | 0.032 | 0.414 |
| ViT | 0.557 | 0.302 | 0.843 | 0.376 | 0.206 | 0.457 | 0.516 | 0.303 | 0.806 | 0.417 | 0.206 | 0.450 |
| ViT In21K | 0.641 | 0.513 | 0.807 | 0.541 | 0.158 | 0.532 | 0.451 | 0.253 | 0.816 | 0.392 | 0.080 | 0.398 |
| ViT-S In21K In1k | 0.515 | 0.362 | 0.828 | 0.700 | 0.145 | 0.510 | 0.512 | 0.301 | 0.830 | 0.441 | 0.161 | 0.449 |
| ViT In21K In1k | 0.405 | 0.245 | 0.759 | 0.319 | 0.058 | 0.357 | 0.425 | 0.215 | 0.784 | 0.393 | 0.061 | 0.376 |
| ViT-L In21K In1k | 0.322 | 0.105 | 0.607 | 0.205 | 0.028 | 0.253 | 0.272 | 0.120 | 0.625 | 0.299 | 0.028 | 0.269 |
| DeiT3 | 0.505 | 0.270 | 0.829 | 0.379 | 0.183 | 0.433 | 0.437 | 0.256 | 0.824 | 0.353 | 0.171 | 0.408 |
| DeiT3 FB In22k In1k | 0.480 | 0.236 | 0.841 | 0.386 | 0.092 | 0.407 | 0.457 | 0.243 | 0.825 | 0.395 | 0.104 | 0.405 |
| DeiT3 In21k In1k | 0.432 | 0.201 | 0.780 | 0.388 | 0.081 | 0.376 | 0.407 | 0.206 | 0.769 | 0.360 | 0.086 | 0.366 |
| DeiT3-L In22k In1k | 0.402 | 0.187 | 0.744 | 0.443 | 0.054 | 0.366 | 0.405 | 0.216 | 0.792 | 0.402 | 0.063 | 0.376 |
| EVA02 | 0.691 | 0.340 | 0.837 | 0.422 | 0.379 | 0.534 | 0.515 | 0.252 | 0.872 | 0.444 | 0.116 | 0.440 |
| EVA02 In1k | 0.418 | 0.186 | 0.800 | 0.323 | 0.153 | 0.376 | 0.431 | 0.226 | 0.876 | 0.363 | 0.118 | 0.403 |
| EVA02 In21k | 0.638 | 0.527 | 0.805 | 0.545 | 0.308 | 0.565 | 0.393 | 0.225 | 0.751 | 0.362 | 0.106 | 0.367 |
| EVA02-S In22k In1k | 0.526 | 0.260 | 0.800 | 0.374 | 0.152 | 0.422 | 0.517 | 0.278 | 0.860 | 0.388 | 0.173 | 0.443 |
| EVA02 In21k In1k | 0.424 | 0.262 | 0.763 | 0.412 | 0.179 | 0.408 | 0.425 | 0.262 | 0.806 | 0.363 | 0.159 | 0.403 |
| EVA02-L In22k In1k | 0.356 | 0.214 | 0.651 | 0.353 | 0.276 | 0.370 | 0.321 | 0.190 | 0.703 | 0.300 | 0.118 | 0.326 |
| ViT CLIP In1k | 0.476 | 0.221 | 0.792 | 0.379 | 0.144 | 0.402 | 0.446 | 0.222 | 0.794 | 0.387 | 0.164 | 0.403 |
| ViT CLIP In12k In1k | 0.379 | 0.190 | 0.659 | 0.357 | 0.091 | 0.335 | 0.373 | 0.157 | 0.693 | 0.331 | 0.071 | 0.325 |
| ViT-L CLIP In12k In1k | 0.320 | 0.167 | 0.583 | 0.375 | 0.038 | 0.297 | 0.299 | 0.159 | 0.631 | 0.326 | 0.047 | 0.292 |

## D.2 CONFORMAL NORMALIZATION NINCO RESULTS

Table 3: False-positive rate (FPR) across models for NINCO dataset using different Mahalanobis variants: **MD\*** uses the empirically optimal $\beta$, **$\hat{\text{MD}}$** uses the regression-predicted $\hat{\beta}$, **MD** (standard) fixes $\beta = 0$, and **MD++** (Mahalanobis++) fixes $\beta = 1$.

| Model | MD* | $\hat{\text{MD}}$ | MD | MD++ | RMD* | $\hat{\text{RMD}}$ | RMD | RMD++ |
|---|---|---|---|---|---|---|---|---|
| BEiTV2 FT In1k | 0.446 | 0.463 | 0.506 | 0.470 | 0.414 | 0.418 | 0.457 | 0.443 |
| BEiTV2 FT In21k | 0.281 | 0.281 | 0.475 | 0.364 | 0.311 | 0.322 | 0.339 | 0.331 |
| DINOV2 | 0.403 | 0.415 | 0.424 | 0.445 | 0.573 | 0.573 | 0.577 | 0.574 |
| DeiT3 | 0.496 | 0.496 | 0.505 | 0.500 | 0.422 | 0.429 | 0.437 | 0.433 |
| DeiT3 FB In22k In1k | 0.455 | 0.464 | 0.480 | 0.455 | 0.445 | 0.455 | 0.457 | 0.445 |
| DeiT3 In21k In1k | 0.423 | 0.426 | 0.432 | 0.423 | 0.349 | 0.349 | 0.407 | 0.387 |
| DeiT3-L In22k In1k | 0.386 | 0.391 | 0.402 | 0.388 | 0.368 | 0.368 | 0.405 | 0.385 |
| EVA02 | 0.535 | 0.630 | 0.691 | 0.630 | 0.499 | 0.515 | 0.515 | 0.523 |
| EVA02 FT In1k | 0.409 | 0.412 | 0.418 | 0.413 | 0.418 | 0.418 | 0.431 | 0.425 |
| EVA02 FT In21k | 0.538 | 0.546 | 0.638 | 0.563 | 0.385 | 0.393 | 0.393 | 0.400 |
| EVA02 FT In21k In1k | 0.406 | 0.406 | 0.424 | 0.407 | 0.387 | 0.388 | 0.425 | 0.406 |
| EVA02-L FT In22k In1k | 0.349 | 0.358 | 0.356 | 0.388 | 0.311 | 0.312 | 0.321 | 0.316 |
| EVA02-S FT In22k In1k | 0.494 | 0.499 | 0.526 | 0.504 | 0.506 | 0.506 | 0.517 | 0.513 |
| MAE FT In1k | 0.481 | 0.485 | 0.516 | 0.488 | 0.428 | 0.433 | 0.463 | 0.450 |
| ViT | 0.553 | 0.557 | 0.557 | 0.556 | 0.511 | 0.513 | 0.516 | 0.514 |
| ViT CLIP In12k In1k | 0.302 | 0.317 | 0.379 | 0.336 | 0.301 | 0.301 | 0.373 | 0.352 |
| ViT CLIP In1k | 0.436 | 0.439 | 0.476 | 0.455 | 0.409 | 0.411 | 0.446 | 0.433 |
| ViT In21K | 0.630 | 0.659 | 0.641 | 0.658 | 0.442 | 0.454 | 0.451 | 0.453 |
| ViT In21K In1k | 0.394 | 0.394 | 0.405 | 0.481 | 0.415 | 0.422 | 0.425 | 0.425 |
| ViT-L CLIP In12k In1k | 0.309 | 0.309 | 0.320 | 0.310 | 0.272 | 0.282 | 0.299 | 0.290 |
| ViT-L In21K In1k | 0.303 | 0.306 | 0.322 | 0.358 | 0.264 | 0.272 | 0.272 | 0.277 |
| ViT-S In21K In1k | 0.499 | 0.500 | 0.515 | 0.505 | 0.512 | 0.514 | 0.512 | 0.514 |

# E    GEOMETRY OF EIGVENVALUES

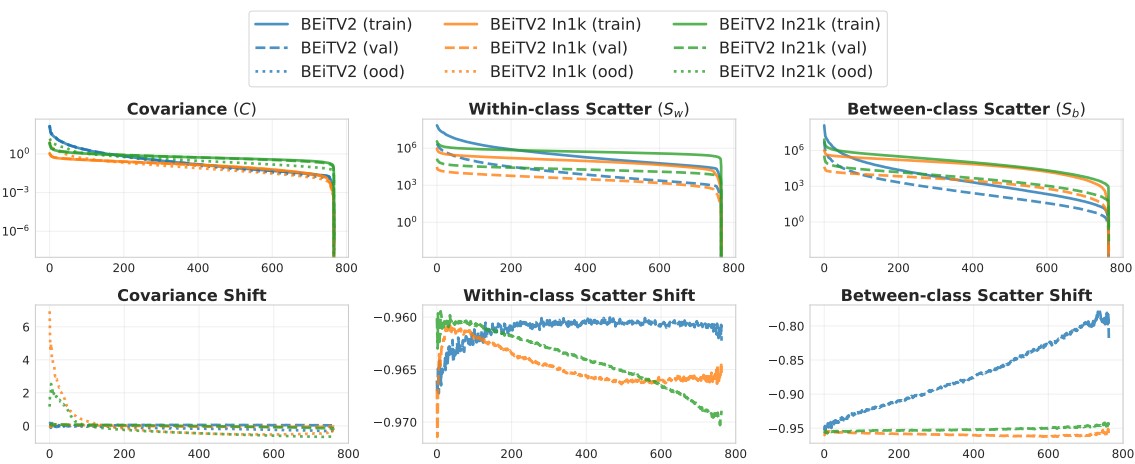

Figure 9:   BEiTV2 eigenspectra and their respective shifts: top—eigenvalues of covariance C, within-class $S_w$, and between-class $S_b$ across train (solid), val (dashed), and OOD (dotted); bottom—corresponding OOD-induced eigenvalue shifts relative to train.

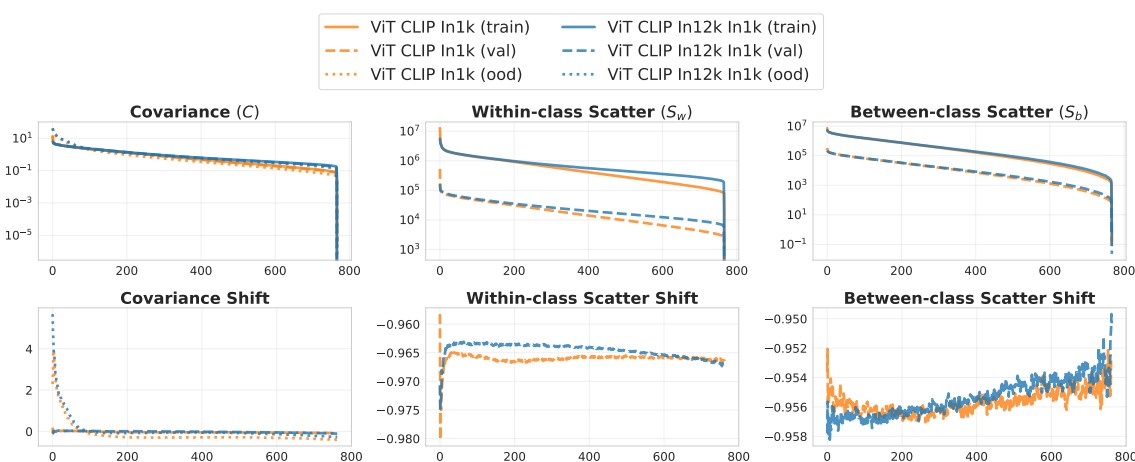

Figure 10:   CLIP eigenspectra and their respective shifts: top—eigenvalues of covariance C, within-class $S_w$, and between-class $S_b$ across train (solid), val (dashed), and OOD (dotted); bottom—corresponding OOD-induced eigenvalue shifts relative to train.

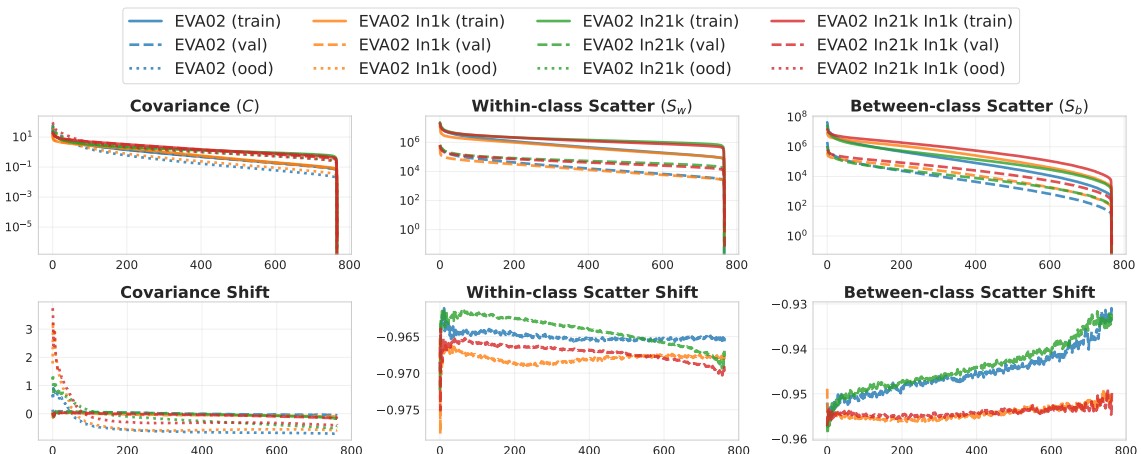

Figure 11: EVA02 eigenspectra and their respective shifts: top—eigenvalues of covariance C, within-class $S_w$, and between-class $S_b$ across train (solid), val (dashed), and OOD (dotted); bottom—corresponding OOD-induced eigenvalue shifts relative to train.

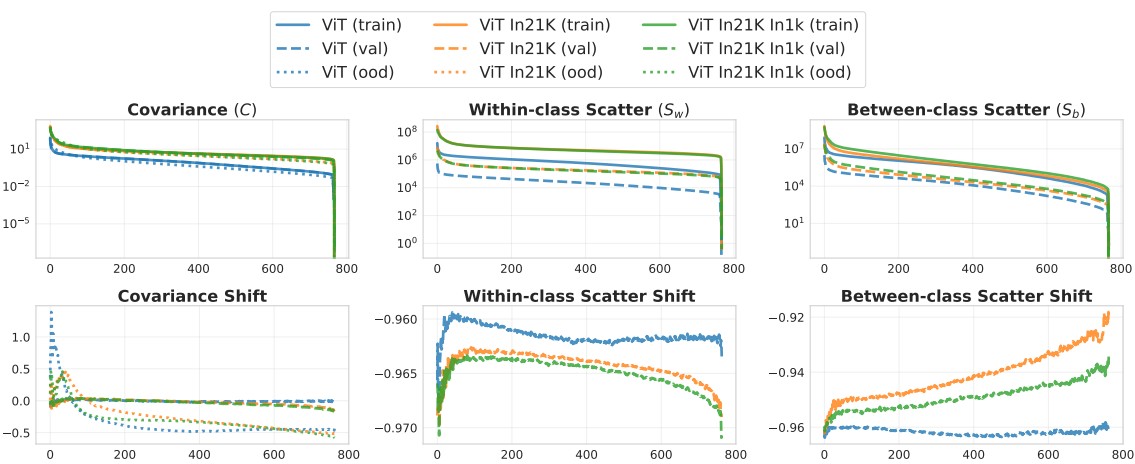

Figure 12: ViT eigenspectra and their respective shifts: top—eigenvalues of covariance C, within-class $S_w$, and between-class $S_b$ across train (solid), val (dashed), and OOD (dotted); bottom—corresponding OOD-induced eigenvalue shifts relative to train.

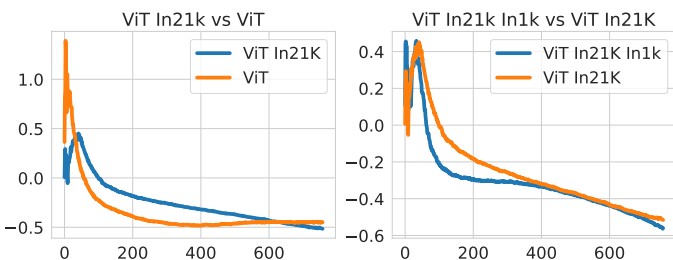

Figure 13: Eigenspectrum of covariance shift between train and OOD data (NINCO) for ViT variants: left—ViT In21K vs ViT; right—ViT In21K In1k vs ViT In21K.

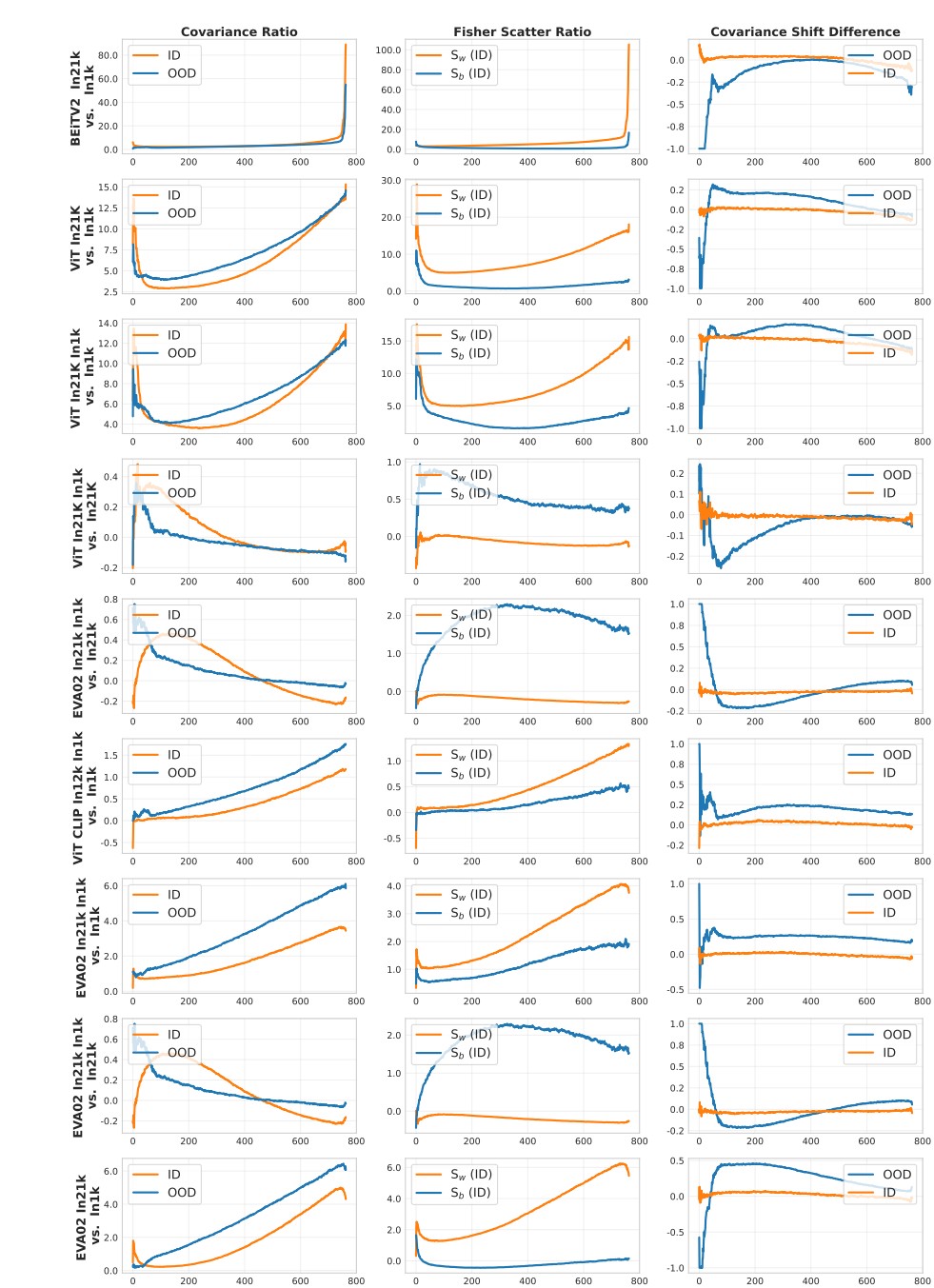

Figure 14: Eigenspectrum differences by model pair (BEiTV2, ViT, EVA02/CLIP): for each pair, we plot ID vs OOD covariance $C$, ID within-class $S_w$ and between-class $S_b$, and covariance-shift curves (OOD and ID), showing relative eigenvalue changes between the first and second model.

## F    Full model names

Table 4: Mapping of model names to checkpoints and sources.

| Model Name | Checkpoint (Version) | Source |
|---|---|---|
| BEiTV2 In1k | beitv2_base_patch16_224.in1k_ft_in1k | timm / huggingface |
| BEiTV2 In21k | beitv2_base_patch16_224.in1k_ft_in22k | timm / huggingface |
| DINOV2 | vit_base_patch14_dinov2.lvd142m | timm / huggingface |
| DINOV3 | dinov3-vitb16-pretrain-lvd1689m | facebook / huggingface |
| MAE In1k | mae_finetuned_vit_base | github.com/facebookresearch/mae |
| ViT | vit_base_patch16_224.augreg_in1k | timm / huggingface |
| ViT In21K | vit_base_patch16_224.augreg_in21k | timm / huggingface |
| ViT In21K In1k | vit_base_patch16_224.augreg_in21k_ft_in1k | timm / huggingface |
| ViT-S In21K In1k | vit_small_patch16_224.augreg_in21k_ft_in1k | timm / huggingface |
| ViT-L In21K In1k | vit_large_patch16_224.augreg_in21k_ft_in1k | timm / huggingface |
| ViT CLIP In1k | vit_base_patch16_clip_224.laion2b_ft_in1k | timm / huggingface |
| ViT CLIP In12k In1k | vit_base_patch16_clip_224.laion2b_ft_in12k_in1k | timm / huggingface |
| ViT-L CLIP In12k In1k | vit_large_patch14_clip_336.laion2b_ft_in12k_in1k | timm / huggingface |
| EVA02 | eva02_base_patch14_224.mim_in22k | timm / huggingface |
| EVA02 In1k | eva02_base_patch14_448.mim_in22k_ft_in1k | timm / huggingface |
| EVA02 In21k | eva02_base_patch14_448.mim_in22k_ft_in22k | timm / huggingface |
| EVA02 In21k In1k | eva02_base_patch14_448.mim_in22k_ft_in22k_in1k | timm / huggingface |
| EVA02-L In22k In1k | eva02_large_patch14_448.mim_m38m_ft_in22k_in1k | timm / huggingface |
| EVA02-S In22k In1k | eva02_small_patch14_336.mim_in22k_ft_in1k | timm / huggingface |
| DeiT3 | deit3_base_patch16_224 | timm / huggingface |
| DeiT3 In21k In1k | deit3_base_patch16_224_in21ft1k | timm / huggingface |
| DeiT3 FB In22k In1k | deit3_base_patch16_384.fb_in22k_ft_in1k | timm / huggingface |
| DeiT3-L In22k In1k | deit3_large_patch16_384.fb_in22k_ft_in1k | timm / huggingface |

## G    Use of AI Assistance

AI assistants, such as ChatGPT, were utilized in various aspects of the research, including coding, data analysis, and writing tasks. These tools helped automate repetitive tasks, generate initial drafts, and assist in exploring potential solutions. However, all AI-generated outputs were reviewed and refined by researchers to ensure accuracy and coherence.

