# OpenReview forum: "Dissecting Mahalanobis: How Feature Geometry and Normalization Shape OOD Detection"
_ICLR.cc/2026/Conference — ICLR 2026 Conference Withdrawn Submission_

### Official Review · Reviewer_3RoL · 2025-10-21

**Soundness:** 3
**Presentation:** 4
**Contribution:** 2
**Rating:** 4
**Confidence:** 4

**Summary:**

The paper benchmarks Mahalanobis-style OOD detectors across modern vision backbones and links OOD performance to representation geometry (using metrics like covariance, within/between class scatter, LID etc.). It then introduces a $\beta$-scaled L2 normalization and trains a leave-one-dataset-out regressor to predict $\beta$ using ID geometry plus auxiliary OOD datasets.

**Strengths:**

- **Clarity.** The text is cleanly written and asks the right questions (I particularly enjoyed the questions on lines ~133, ~197), leading readers to an intuitive and informative analysis.

- **Per dimension contribution $S_i$.** The view of per-PC contributions is useful and well-motivated. Authors may consider calculating the FPR95 with the ith principal component. Aggregating the first k is in my opinion harder to read and $S_i$ as a metric could be misleading as it is a distributional metric. However, we do the OOD detection on the sample level, $S_i$ would be more appropriate if we were tackling set level OOD detection.

- **Predictive diagnostics.** Geometry-based metrics (e.g. $LID \cdot S_w$) correlate with OOD performance and feel like the right direction to explore when a backbone is suitable for post-hoc OOD.

- **Spectral-shift observation.** The link between OOD-induced spectral shift and pretraining dataset size is a practically important observation. Here one get insights on how robust we expect the models to be on the OOD representations.

Overall, the first half builds a strong understanding and experiments are coherent and in line with the insights.

**Weaknesses:**

- **Fairness of comparisons.** This is the biggest concern I have. The $\beta$-regressor trained with auxiliary OOD which moves the method away from training-free post-hoc and makes direct comparisons to strictly post-hoc Mahalanobis++ less fair.
- **Baseline gap.** Natural follow up to my previous point is that if auxiliary OOD is allowed, a simple linear OOD classifier can be put as a baseline to let us understand the value add of $\beta$ scaling.
- **Section bridge.** The normalization section feels disconnected from the diagnostics. A natural bridge would be using the predictive metric ($LID \cdot S_w$) to set $\beta$ without auxiliary OOD, keeping the method post-hoc and strengthening the narrative.
- **Maha++ missing in sections 3 and 4.** Mahalanobis++ should be present in the core geometry analyses (not just final comparisons) to catch potential insights from normalized feature space.

**Questions:**

- **Clarification on contribution 3 (line 66).** "Predicting optimal $\beta$ using only ID data" is very misleading. Please explicitly state that auxiliary OOD datasets are used to train the $\beta$-regressor. Using only ID data implies you use nothing else than what model is trained.

- **Per-PC signal (line 186).** You observe smaller variance directions can be more discriminative for OOD. It feels very counterintuitive to me, do you have an intuition that explains this behavior in some models?

- **$S_b / S_w$ ratio (line 231).** Adding a side by side plot of the ratio vs OOD performance would support these claims. Right now it is not straightforward to me the conclusion that it would favor OOD detections directly. As written, the ratio alone only tells how the model assigns ID points to the representation space. Therefore, it doesn’t guarantee separability for OOD; an empirical correlation would help.

- **Benchmark breakdowns.** I would enjoy seeing near vs far OOD breakdowns to match the analyses done in the literature more easily. Moreover, the varying performances over near and far cases potentially reveal additional insights on the representation behaviors of these models.


### Minor Comments/Edits

- **Line 83.** The sentence beginning "However, our study reveals..." would read better if you immediately state how for a better reading experience.

- **Line 145 & 149.** These insights have been observed before; please cite [1] and [2] again at these points to orient readers.

- **Start of the Section 4 (line 194).** Up to here, RMD seems consistently ahead on FPR metrics. I already made my comment on how average separability metric could be misleading, I would temper the claim here.


[1] Bitterwolf, J., Mueller, M., & Hein, M. (2023). In or out? fixing imagenet out-of-distribution detection evaluation. arXiv preprint arXiv:2306.00826.

[2] Ren, J., Fort, S., Liu, J., Roy, A. G., Padhy, S., & Lakshminarayanan, B. (2021). A simple fix to mahalanobis distance for improving near-ood detection. arXiv preprint arXiv:2106.09022.

---

### Official Review · Reviewer_NA6z · 2025-10-23

**Soundness:** 2
**Presentation:** 1
**Contribution:** 2
**Rating:** 4
**Confidence:** 4

**Summary:**

The paper investigates the geometric properties of neural network feature representations that influence the success or failure of out-of-distribution (OOD) detection, focusing on variants of Mahalanobis-based detectors. The main contributions are: (1) empirical analyses demonstrating how the performance of several Mahalanobis detectors correlates with various spectral metrics in feature space, and (2) a proposed feature normalization method, given by $\phi(z) = z / \||z\||^\beta$, where the parameter $\beta$ is fitted to improve the performance of Mahalanobis detectors.

**Strengths:**

**(S1) Analysis of factors underlying successful OOD detection:** The paper examines which properties of neural network representations geometry lead to successful OOD detection using Mahalanobis-based detectors. This is an important and timely topic, as OOD detection performance is known to vary considerably across settings, and concrete insights or recommendations on what drives success remain scarce in the literature. The paper provides empirical evidence on how various metrics characterizing the feature distribution influence the performance of Mahalanobis OOD detectors.


**(S2) Scope of empirics:** The empirical analysis covers a wide range of models and spectral metrics.

**Weaknesses:**

**(W1) Clarity and precision:** The paper’s precise contributions are difficult to understand due to clarity issues. For example:
- The term “features” is never clearly defined. In Mahalanobis OOD detectors, features typically correspond to activations from specific hidden layers, but performance can depend strongly on which layers are used and how they are aggregated. The paper does not specify the exact setup considered.
- Several quantities presented in Figure 4 are not formally defined and cannot be easily inferred from the text. For instance, the “Fisher scatter ratio” usually involves both $S_w$ and $S_b$, yet the corresponding plot separates them into distinct lines without explanation.
- Most figures lack x-axis labels, which makes them difficult to understand.

**(W2) Method for fitting $\beta$:** The proposed normalization method requires fitting the parameter $\beta$. *Contrary to the claim made in the contributions summary*, the procedure for choosing $\beta$ *requires access to OOD data* (the authors propose to train on *all OOD datasets except the target one*). Moreover, the parameter must be refitted for each setting, as Figure 7 shows that its optimal value varies widely across models and datasets. Given this variability, the choice of OOD data used for fitting $\beta$ could substantially influence the results. This raises concerns about the practical applicability of the proposed method, since a comprehensive OOD dataset is usually not available in practical settings.

**(W3) Insight and discussion:** To my mind, the biggest weakness is that the paper presents many isolated results that are not synthesized into a coherent framework. The first part of the contributions shows that Mahalanobis OOD detection depends on various spectral metrics of the feature space. The second part proposes a normalization technique with a tunable parameter $\beta$ to improve performance. However, the paper does not discuss how this normalization affects the spectral properties relevant to OOD detection, or why fitting $\beta$ should modify these properties in a favorable way. If I am not missing something, these questions are not addressed in the discussion, and no formal results are provided to support the claim that normalization makes the geometry “better.” Therefore, the proposed method’s connection to the geometric arguments presented in the paper seems unsupported.

**(W4, minor) Scope limited to Mahalanobis detectors:** Since Mahalanobis detectors rely on fitting Gaussian distributions to ID data, it is expected that spectral properties of the feature space correlate with OOD detection performance. It would be more interesting and impactful to investigate which factors influence the success of OOD detection methods more generally, beyond the Mahalanobis family.

**Questions:**

- Could the authors explicitly define what is meant by “features”, and how they expect the praticular choice of representations to affect the results?
- How does the proposed normalization affect the geometric properties that the paper identifies as important for OOD detection?

---

### Official Review · Reviewer_AKAg · 2025-10-31

**Soundness:** 3
**Presentation:** 3
**Contribution:** 3
**Rating:** 6
**Confidence:** 3

**Summary:**

This paper conducted a comprehensive empirical study
across diverse image foundation models, datasets, and distance normalization schemes.
First, the study shows that Mahalanobis-based methods aren’t universally reliable. Second, the authors define the ideal geometry for data representations and demonstrate that spectral
and intrinsic-dimensionality metrics can accurately predict a model’s out-of-distribution
(OOD) performance. Finally, the paper analyzes how normalization impacts OOD performance.
Building upon these studies, a conformal generalization of
ℓ2 normalization that allows to control the degree of radial expansion of the representations geometry, which in turn helps improve OOD detection, is proposed.

**Strengths:**

1. A comparative study of several Mahalanobis-based OOD detectors across diverse models, including
a per-dimension analysis of its variants.
2. An analysis revealing that OOD performance is strongly predicted by the in-distribution geometry
of features, such as spectral decay and intrinsic rank.
3. The introduction of β-scaled ℓ2 normalization and a method to predict the optimal β using only
in-distribution data.

**Weaknesses:**

1. Why does the combination of the LID and  S_w metrics outperform single metrics (e.g., using LID or entropy alone)? Visualizing the feature clustering distributions of different models  may intuitively illustrate the relationship between this combined metric and clustering quality.

2. Lack of comparison with current state-of-the-art (SOTA) methods. The paper only compares against traditional Mahalanobis-distance variants, but omits recent SOTA OOD detection approaches based on “feature calibration” or “contrastive learning.” This makes it difficult to position the proposed method within the current research landscape.

3. Experiments are limited to image classification models and do not validate the method’s effectiveness on downstream tasks such as object detection or segmentation. Adding OOD detection experiments on 1–2 object detection models (e.g., Faster R-CNN) would better demonstrate the method’s task adaptability.

**Questions:**

Please refer to the Weaknesses.

---

### Official Review · Reviewer_FWsG · 2025-10-31

**Soundness:** 3
**Presentation:** 4
**Contribution:** 3
**Rating:** 4
**Confidence:** 4

**Summary:**

This paper investigates the factors that determine the effectiveness of **Mahalanobis-based Out-of-Distribution (OOD) detectors**. Through a comprehensive empirical study across various vision foundation models, the authors demonstrate that the performance of these detectors is not universal  and is often weakly correlated with a model's in-distribution (ID) classification accuracy.

The core argument is that OOD detection performance is primarily driven by the **geometric properties of the in-distribution feature space**. The study analyzes the spectral properties of the feature covariance ($C$), within-class scatter ($S_w$), and between-class scatter ($S_b$) matrices. A key finding is that an "ideal geometry" for OOD detection involves a balance between **local manifold complexity** (measured by Local Intrinsic Dimensionality, LID) and **intra-class compactness** (measured by the spectral slope of $S_w$). The authors show that the product of these two metrics serves as a strong predictor of a model's OOD performance.

Building on this, the paper examines the role of **feature normalization**. It argues that deep features often violate the Gaussian assumptions of the Mahalanobis distance  and introduces a novel **conformal $l_2$ normalization** to reshape the feature space geometry. This method provides a "control knob" that generalizes prior work and can be optimized to improve OOD detection.

**Strengths:**

The paper introduces a novel $\beta$-scaled (conformal) $l_2$ normalization. This is a creative generalization of prior work, reframing standard Mahalanobis ($\beta=0$) and Mahalanobis++ ($\beta=1$) as specific points in a continuous space of geometric transformations.

This "control knob" for feature geometry is a new and elegant contribution.The identification of the product of Local Intrinsic Dimensionality (LID) and the $S_w$ (within-class scatter) slope as a strong predictor of OOD performance is highly original .

Most prior work correlates performance with single metrics, whereas this combined metric uniquely captures the trade-off between manifold complexity and cluster compactness, which the authors define as the "ideal geometry" .
The problem formulation of predicting the optimal normalization parameter $\hat{\beta}$ using a regression model trained only on in-distribution geometry and auxiliary OOD datasets is a novel and practical approach to the problem of hyperparameter tuning without target OOD data

The finding that OOD performance is predictable from in-distribution geometric properties (LID $\times$ $S_w$ slope) is a fundamental contribution that could guide future work in representation learning, suggesting that models can be explicitly trained to have an "ideal geometry" for OOD robustness.

**Weaknesses:**

- Missing some related work analyzing the mahalanobis distance too

for example:

1-  Kamoi, Ryo, and Kei Kobayashi. “Why Is the Mahalanobis Distance Effective for Anomaly Detection?” arXiv:2003.00402, arXiv, 30 Apr. 2020. arXiv.org, http://arxiv.org/abs/2003.00402.

2- Tajwar, Fahim, et al. "No true state-of-the-art? ood detection methods are inconsistent across datasets." arXiv preprint arXiv:2109.05554 (2021).

- Per dimension analysis of Mahalanobis distance is similar to the analysis conducted in [1].

- Dependence on Auxiliary OOD Data: The paper's main practical solution, the regression framework for predicting the optimal $\beta$, is not fully "post-hoc" in the strictest sense. It relies on a "Leave-One-Dataset-Out" scheme, meaning it must be trained using a collection of other OOD datasets to learn the relationship between in-distribution geometry and the optimal $\beta$

- Limited Scope of Experimental Comparison: The evaluation of the proposed method is not benchmarked against the current state-of-the-art. While the paper uses the OpenOOD protocol , the results tables only compare variants of Mahalanobis distance (MD, RMD) and report a single metric (FPR@95). A more convincing evaluation would include a comprehensive table comparing the proposed optimized detector ($\phi_{\beta}(z^{\prime})$) against a wider range of non-Mahalanobis baselines on the OpenOOD benchmark, using standard metrics like AUROC in addition to FPR.

**Questions:**

- The per-dimension analysis in Section 3.3  appears quite similar to the analysis in Kamoi & Kobayashi (2020)[1], which is not cited. Could the authors clarify the novel insights from their analysis beyond this prior work?

-  The "ideal geometry" metric (LID $\times$ $S_w$ slope) is shown to be a strong correlational predictor of OOD performance. Have the authors considered or investigated a causal link? For example, have you explored using this metric as a regularization term during model training to actively enforce this "ideal geometry," and if so, does it improve downstream OOD detection?

References:
1- Kamoi, Ryo, and Kei Kobayashi. “Why Is the Mahalanobis Distance Effective for Anomaly Detection?” arXiv:2003.00402, arXiv, 30 Apr. 2020. arXiv.org, http://arxiv.org/abs/2003.00402.

---

### Official Review · Reviewer_zoif · 2025-11-01

**Soundness:** 3
**Presentation:** 3
**Contribution:** 3
**Rating:** 6
**Confidence:** 4

**Summary:**

This paper investigates why Mahalanobis-based out-of-distribution (OOD) detection methods perform inconsistently across different vision foundation models. Through extensive empirical analysis, the authors show that OOD performance strongly correlates with geometric properties of in-distribution (ID) feature spaces, in particular intrinsic dimensionality and within-class scatter structure. They propose a $\beta$-scaled $\ell^2$ normalization, a conformal transformation that radially reshapes feature geometry to better align with Mahalanobis assumptions or enhance OOD separation. A regression model trained on ID geometry metrics and auxiliary OOD data can predict a near-optimal $\beta$ without access to target OOD samples, improving detection across diverse models and datasets.

**Strengths:**

The main strengths of the paper are listed below:

* Comprehensive empirical study: The paper evaluates a wide range of modern self-supervised and foundation models across multiple OOD benchmarks.

* Insightful geometric analysis: The paper identifies that the product of Local Intrinsic Dimensionality (LID) and within-class scatter slope strongly predicts OOD performance (Spearman $\rho$ up to 0.85).

* Practical contribution: The paper introduces $\beta$-scaled $\ell^2 normalization, a simple, effective, and theoretically grounded method to control feature geometry.

* Data-efficient tuning: The paper demonstrates that optimal $\beta$ can be predicted using only ID data and auxiliary OOD datasets, avoiding reliance on target OOD samples.

**Weaknesses:**

The paper can be improved along the following lines:

* Limited theoretical grounding: While empirical correlations are strong, the paper lacks formal theoretical justification for why the LID × slope product is predictive.

* Dependence on auxiliary OOD data: The regression-based $\beta prediction still requires access to other OOD datasets (via leave-one-out), which may not always be available in real-world settings.

* Narrow scope of normalization: The paper focuses only on radial (conformal) transformations. Could other geometric manipulations such as anisotropic scaling be explored?

* Computational overhead: Fitting class-conditional Gaussians and performing eigen-decompositions may be costly for very large models or high-dimensional features. This limits the applicability of this study to (only?) small models.

**Questions:**

Overall the paper needs to be more theoretically grounding. In the absence of that, could the author justify their focus on radial transformations and not on other geometric manipulations such as anisotropic scaling?

---

### Official Review · Reviewer_DQiN · 2025-11-04

**Soundness:** 2
**Presentation:** 2
**Contribution:** 2
**Rating:** 2
**Confidence:** 5

**Summary:**

This paper explores how the geometry of learned representations and feature normalisation affect the performance of Mahalanobis-based out-of-distribution (OOD) detection methods. The authors conduct a comprehensive empirical study across a range of self-supervised and pretrained vision models, examining spectral and manifold properties of feature spaces. They introduce a \beta-scaled conformal \ell-2 normalisation to control radial feature geometry and propose a regression-based approach to predict the optimal \beta value using only in-distribution (ID) data. The stated goal is to provide both an explanatory framework linking representation geometry and OOD performance, and a practical technique for improving Mahalanobis-based detection.

**Strengths:**

The paper presents an empirical analysis that covers a set of foundation models, datasets, and normalisation schemes. This breadth of experimentation offers comparative insights into how representation learning choices influence OOD detection ability. The geometric viewpoint that relates intrinsic dimensionality, eigenvalue spectra, and normalisation effects to OOD reliability is conceptually appealing and shows some analytical depth. The authors’ systematic investigation of covariance, within-class and between-class scatter, and their spectral properties demonstrates a commendable effort to interpret model behaviour beyond raw performance metrics. Moreover, the proposed \beta-scaled normalisation is a simple, easily reproducible idea that gives practitioners a tunable parameter, if handled well, for shaping feature geometry. The attempt to predict this parameter through regression gives some hint to step towards automating hyperparameter tuning for post-hoc OOD detectors.

**Weaknesses:**

Although the study is broad and the analyses are detailed, the technical depth of the contribution remains limited. Several major concerns are below.

The first major concern is that the theoretical foundation of the work is a bit shaky. There is an obvious disconnection between OOD and optimality. Each test, based on FPR@95 as a target and surrogate for OOD, appears to be a single trial without repeat. How to justify the findings are robust but not random? There are many hand-wavy statements indicating "better for OOD detection" without any source of evidence, including many tests that are highly training dependent (for example, suboptimal due to insufficient training) or sampling rate (sufficiency of the data). As an example of the arbitrary choice in this paper, the proposed transformation is introduced as a geometric intuition but is not supported by a rigorous derivation showing why it improves OOD separability. The Riemannian interpretation provided in Appendix C is ok but not connected to any probabilistic or optimisation principle that would justify its efficacy. As a result, the paper remains largely empirical rather than theoretically grounded.

Second is that the novelty of the work is modest. The \beta-scaling formulation extends the idea of \ell_2 normalisation in Mahalanobis++ (Mueller & Hein, 2025) but offers little conceptual advance beyond introducing a continuous scaling factor. Similarly, the regression model predicting \beta is only a simple mapping from empirical metrics to a hyperparameter and does not introduce new methodological insights. Consequently, the paper reads more as a diagnostic study than as a genuine algorithmic contribution.

A third issue is that the empirical results do not convincingly demonstrate causal relations between geometry and OOD performance. Most conclusions are based on correlations rather than controlled experiments. The claim that representation geometry “drives” OOD detection is not substantiated, as confounding factors such as model size, dataset diversity, or training objective are not accounted for. Without controlled ablations, it is impossible to determine whether geometry itself or these associated factors explain the observed trends.

Another major concern is the limited specification and weak predictive performance of the regression model for \beta. Important implementation details, including the regression type, feature preprocessing, and regularisation, are omitted. The reported R2 values (0.25 for MD and 0.47 for RMD) are low, indicating that the predictor explains little variance. Figure 8 shows large deviations from the diagonal, yet this is not critically discussed. The claim that the regression achieves near-oracle performance is therefore overstated.

The evaluation protocol lacks clarity and statistical rigour. The process for selecting the “empirical optimal” \beta is underdefined, and it is unclear whether results are averaged across seeds or if statistical confidence is reported. In Table 1, the differences between methods are small and presented without variance estimates. This weakens the validity of the claimed improvements.

The interpretability and necessity of the various geometric metrics also remain questionable. The study introduces a large set of spectral and manifold statistics, but many are highly correlated and redundant. The proposed composite metric (the product of Local Intrinsic Dimensionality and within-class scatter slope) appears to have been found post-hoc and is not compared systematically against simpler alternatives. The reported Spearman correlation of 0.85 is presented as strong evidence but is not verified across dataset splits.

**Questions:**

The authors have started a good empirical study that sheds light on how feature geometry and normalisation influence Mahalanobis-based OOD detection. The findings are partially informative, but the paper’s principal claims exceed what is demonstrated. The regression model for predicting \beta using other data sets' statistics to predict a remaining one seems strange. Also it shows weak generalisation. To strengthen this work, future versions should (1) establish a formal link between the conformal transformation and probabilistic assumptions of the Mahalanobis metric, (2) provide robust and statistically validated evaluation protocols, and (3) strengthen the prediction model for \beta considering the real world scenarios for practitioners who may have a single data set at hand. With these additions, the paper could become a valuable contribution to the understanding of representation geometry in OOD detection.

---

### Official Review · Reviewer_N2nU · 2025-11-12

**Soundness:** 3
**Presentation:** 3
**Contribution:** 2
**Rating:** 4
**Confidence:** 4

**Summary:**

This paper presents a comprehensive empirical study of Mahalanobis distance-based out-of-distribution (OOD) detection methods, with a focus on understanding how the feature representation geometry and normalization techniques influence OOD detection performance. The authors evaluate a diverse collection of modern vision models (self-supervised and pretrained transformers) across multiple benchmarks, revealing that Mahalanobis-style detectors exhibit highly variable effectiveness across models and settings. A key finding is that intrinsic properties of the in-distribution feature space, such as the spectral decay of covariance eigenvalues and estimates of intrinsic dimensionality, strongly correlate with OOD detection success. Building on these insights, the paper introduces a conformal $\ell_2$ normalization technique parameterized by $\beta$, which allows one to contract or expand the radial distribution of feature vectors. By tuning $\beta$, the representation’s geometry can be adjusted to better align with the Mahalanobis detector’s Gaussian assumptions, thereby improving OOD detection performance. Importantly, the paper proposes a simple regression-based approach to predict the optimal $\beta$ using only in-distribution data, without requiring any OOD samples. This learned predictor uses metrics derived from the training features (e.g. intrinsic dimensionality, within-class scatter) to estimate the best $\beta$, and achieves nearly the same OOD detection performance as an oracle that had access to OOD validation data.

**Strengths:**

- The study is comprehensive in scope, evaluating Mahalanobis-type OOD detectors across a wide range of foundation vision models, training regimes, and datasets. The authors follow established protocols (e.g. OpenOOD) and test on multiple OOD benchmark suites (NINCO, iNaturalist, OpenImages-O, Textures, etc.), lending credibility to the findings. This broad evaluation reveals interesting trends, e.g., the Relative Mahalanobis Distance (RMD) consistently outperforms the standard Mahalanobis scoring, especially for models that were pretrained on large data but not fine-tuned on the target domain.
- The paper provides a deep analysis connecting representation geometry to OOD performance, which appears original and insightful. The authors identify measurable statistics of the feature space (covariance eigenvalue distribution, intrinsic dimensionality (ID), within-class scatter, etc.) and show that these correlate strongly with how well OOD examples can be detected. Notably, they find that a compound metric, the product of Local Intrinsic Dimensionality and scatter slope, is highly predictive of a model’s OOD detection effectiveness.
- Building on the above geometric insights, the paper introduces $\beta$-scaled $\ell_2$ normalization as a novel extension of prior Mahalanobis-based methods. Whereas earlier work (e.g., Mahalanobis++ by Mueller & Hein, 2025) simply applied unit-length normalization to features (equivalent to a fixed $\beta = 1$ case), here the authors generalize this idea by allowing a continuous tuning of the feature norms via the $\beta$ parameter. This conformal transformation of the feature space is an original contribution: it provides a direct “knob” to control the radial spread of feature vectors, effectively interpolating between the original Euclidean feature space ($\beta = 0$, no normalization) and a fully normalized hypersphere ($\beta = 1$), and even beyond ($\beta > 1$ for extra contraction, or $\beta < 0$ for expansion). The paper demonstrates that by choosing an appropriate $\beta$ for a given model, one can significantly improve OOD detection performance (by tightening clusters to better fit Gaussian assumptions or loosening them if needed).
- The paper is well-written and structured in a logical manner. It clearly lays out the problems and questions in early sections, follows up with corresponding analyses, and then transitions into the proposed method. The use of visuals (plots of OOD separation per dimension, conceptual diagram of $\beta$-scaling effects, etc.) is effective in reinforcing the arguments. Important claims are either theoretically explained or empirically demonstrated. The authors do a commendable job connecting the dots between observations (e.g. a model’s spectral properties) and the design of their method ($\beta$ normalization), which makes the paper easy to follow.

**Weaknesses:**

- **Limited comparison to non-mahalanobis baselines**: A notable limitation is that the study restricts its focus to Mahalanobis-based detectors (standard MD, Relative MD, etc.) and does not compare against other mainstream OOD detection approaches. Methods like the Maximum Softmax Probability (MSP) baseline or energy-based scores are standard references in OOD detection literature; however, the paper does not report results for these alternatives. This omission makes it difficult to gauge the absolute improvement or significance of Mahalanobis-based methods in comparison to simpler or better-known baselines. For example, if a simpler MSP or energy score performs on par with the tuned Mahalanobis detector, that would affect the paper’s claim of advancing the state-of-the-art. Originality and significance could be better established by demonstrating that the proposed techniques outperform not only variants of Mahalanobis, but also other competitive OOD detection methods.
- **Primarily empirical findings without deeper theoretical explanation**: The connection drawn between feature geometry metrics and OOD performance, while intriguing, is based on empirical correlation rather than a derived theory. The paper shows strong correlations (e.g., between the product of LID and scatter slope and the detector’s FPR@95), but it stops short of providing a theoretical understanding of why these particular metrics are predictive. As a result, the insight, though useful, can feel a bit post hoc. A reader might wonder: to what extent are these geometric properties causally influencing OOD detection capability, versus simply being correlated indicators?
- **Incremental methodological novelty**: The proposed $\beta$-scaled $\ell_2$ normalization, though useful, could be perceived as a relatively incremental extension over prior art. The idea of normalizing feature vectors ($\ell_2$ normalization) to improve Mahalanobis detection was already introduced in earlier work (referred to as Mahalanobis++, which corresponds to the special case $\beta = 1$). This paper extends that concept by making the normalization strength tunable via $\beta$, which is a natural generalization. While this generalization is well-motivated and yields empirical gains, it may not represent a fundamentally new direction in OOD detection but rather a refinement of an existing technique.
- **Dependency on auxiliary OOD data for $\beta$ tuning**: Although the authors stress that their regression model selects $\beta$ “without access to target OOD data,” in practice the training of this regression does rely on OOD datasets (just not the target one). The experimental setup uses a meta-dataset of OOD experiences: the regression is trained using several auxiliary OOD datasets (and corresponding in-distribution features) to learn a mapping from feature statistics to optimal $\beta$. This implies that the method’s success partly hinges on having a representative set of OOD scenarios available beforehand.
- **Presentation issues**: (1) Most of the figures and tables in the paper are a bit too small (the text are difficult to read). (2) Inconsistent capitalization in paragraph titles, e.g., “Key Insights” and “Spectral ratios”.

**Questions:**

- How does the $\beta$-scaled Mahalanobis detector compare to other common OOD detection methods (e.g., MSP, ODIN, energy-based scores) on the same benchmarks?
- What data and procedure were used to train the $\beta$ prediction regression model, and how confident can we be in its generalization to arbitrary new OOD scenarios? In particular, if one encounters a novel OOD setting without any related auxiliary datasets, can the regression still be applied (perhaps using your pre-trained model)?

---

### Note · Authors · 2025-11-23

I have read and agree with the venue's withdrawal policy on behalf of myself and my co-authors.